# Performance Comparison of Two In-House PCR Methods for Detecting *Neisseria meningitidis* in Asymptomatic Carriers and Antimicrobial Resistance Profiling

**DOI:** 10.3390/diagnostics15050637

**Published:** 2025-03-06

**Authors:** Mekonnen Atimew, Melaku Yidenekachew, Marchegn Yimer, Ashenafi Alemu, Dawit Hailu Alemayehu, Tadelo Wondimagegn, Fitsumbiran Tajebe, Gashaw Adane, Tesfaye Gelanew, Getachew Tesfaye Beyene

**Affiliations:** 1Armauer Hansen Research Institute (AHRI), Addis Ababa P.O. Box 1005, Ethiopiadawit.hailu@ahri.gov.et (D.H.A.); tesfaye.gelanew@ahri.gov.et (T.G.); 2Department of Medical Laboratory Sciences, College of Medicine & Health Sciences, Wolaita Sodo University, Wolaita Sodo P.O. Box 138, Ethiopia; 3Department of Immunology and Molecular Biology, College of Medicine & Health Sciences, University of Gondar, Gondar P.O. Box 196, Ethiopia

**Keywords:** PCR, *Neisseria meningitidis*, pharyngeal swabs, *sodC*, *ctrA*, antimicrobial resistance, Ethiopia

## Abstract

**Background/Objective:** Bacteriological culture has been a widely used method for the detection of meningococcus, but it has low sensitivity and a long turnaround time. Molecular detection targeting capsule transport A (*ctrA)* gene has been used, but over 16% of meningococcal carriage isolates lack *ctrA*, resulting in false-negative reports. The Cu-Zn superoxide dismutase gene (*sodC*) is specific to *N. meningitidis*, and is not found in other *Neisseria species*, making it a useful target for improved detection of non-groupable meningococci without intact *ctrA*. The primary objective of this study was to evaluate the performance compassion of two in-house PCR methods, *sodC* gene- and *ctrA* gene-based PCR assays, for detecting *N. meningitidis* in asymptomatic carriers. The secondary objective was to assess antimicrobial resistance profiling of *N. meningitidis* isolates. **Methods:** The performance of *sodC* gene-based PCR assay compared to *ctrA* gene-based PCR for detection of *N. meningitidis* was evaluated using clinical samples (pharyngeal swabs; *n* = 137) collected from suspected asymptomatic carriers and culture-confirmed meningococci isolates (*n* = 49). Additionally, the antimicrobial sensitivity of the 49 isolates against antimicrobial drugs was determined using a disk diffusion test. **Result:** Of 49 DNA samples from culture-positive *N. meningitidis* isolates, the *sodC* gene-based PCR accurately identified all 49, whereas the *ctrA* gene-based PCR identified only 33 out of 49. Of 137 pharyngeal swabs, the *sodC* gene-based assay detected *N. meningitidis* DNA in 105 (76.6%), while the *ctrA*-based assay detected *N. meningitidis* DNA in 64 (46.7%). Out of the 49 N. meningitidis isolates, 43 (87.8%) were resistant to amoxicillin, 42 (83.7%) to ampicillin, 32 (65.3%) to trimethoprim–sulfamethoxazole, 22 (44.9%) to ceftazidime, 18 (36.7%) to ceftriaxone, and 7 (15.2%) to meropenem. Additionally, the majority of the isolates, 36 (73.5%), were sensitive to cefepime, 31 (63.3%) to ceftriaxone and meropenem, and 26 (53.1%) to ceftazidime. **Conclusions**: The findings of this study highlight the necessity of adopting non-capsular *sodC*-based PCR to replace *ctrA* in resource-constrained laboratories to improve *N. meningitidis* detection, and underscore the importance of periodic antimicrobial resistance surveillance to inform and adapt treatment strategies.

## 1. Introduction

*Neisseria meningitidis*, often referred to as meningococcus, is a Gram-negative bacterium that can cause a spectrum of diseases ranging from mild sepsis with rapid recovery, to fulminant meningococcemia [1]. It is carried by around 5–10% of healthy individuals in the nasopharynx [2]. Every year, more than 1.2 million cases of bacterial meningitis are estimated to occur globally, causing about 14.25% of deaths [3]. In Ethiopia, bacterial meningitis is an important cause of premature death and disability, being the 9th most common cause of years of life lost and disability-adjusted life years. Early treatment is essential in the clinical management of meningitis. A delay in therapy negatively affects the prognosis for patients with meningitis [4]. The incidence of meningitis varies depending on age, geographic location, species, genotype, strain, and serotype of the causative agents [5].

Accurate diagnosis and early treatment of meningococcal infections are essential due to their worldwide distribution, increased case fatality and morbidity rate, epidemic potential, and the serious complications that can occur [6]. For confirming the etiology, CSF and/or blood culture have been used as the gold standard for the diagnosis of meningococcal infection [7]. Identification and detection of bacterial pathogens in cases of suspected meningitis are important in guiding appropriate treatment and prophylaxis. However, diagnosis of bacterial meningitis is often difficult [8]. Traditional laboratory diagnosis of meningococcal disease (MD) has relied heavily on bacteriologic culture methods, but the bacterial growth rates, particularly in patients who have received pre-admission antibiotic treatment, are very low and the culture methods have low sensitivity due to the frequent initiation of antimicrobial therapy before clinical sample collection [9]. Many studies have shown that the high rates of morbidity and mortality linked to MD, particularly in children, are partly caused by delayed detection and diagnosis [10,11,12,13]. 

Molecular assays can be used to diagnose invasive meningococcal infections when previous antibiotic therapy may inhibit bacterial growth [7]. The capsule transport operon A (*ctrA)* gene is the most commonly targeted gene for molecular detection of *N. meningitidis* because *ctrA* is thought to be present in all invasive strains of *N. meningitidis* due to the importance of the capsule in preventing complement-mediated killing [14]. However, the high genetic diversity of *N. meningitidis* and rearrangements in the *ctrA* gene make the detection of this important pathogen by *ctrA*-targeted PCR testing difficult [15,16]. Studies show that *ctrA* is absent in 16% or more of carriage isolates [17,18]. ctrA-based assays have been shown to produce false-negative results due to *ctrA* sequence variations, which commonly result from multiple nucleotide substitutions [19,20,21]. Several cases of invasive and sometimes fatal disease caused by capsule-null (cnl) strains have been reported to lack *ctrA* [22,23,24].

Taken together, these findings question the utility of the *ctrA*-based PCR assay for the detection of *N. meningitis* in carriage specimens and warrant the need to develop reliable molecular approaches for the detection and characterization of this pathogen. Multiplex PCR targeting non-capsular genes such as the Cu-Zn superoxide dismutase gene (sodC), the phenol metabolism gene *(metA*), and the sulfite exporter (*tauE*) could be alternative or complementary targets to *ctrA* to improve the detection of *N. meningitidis* in carriage specimens [25]. Specifically, molecular methods targeting only the *sodC* gene-based PCR assay may be cost-effective diagnostic methods in resource-constrained settings, and have several advantages over culture-based biochemical assays, including increased sensitivity, specificity, speed, and efficiency. In support of its specificity to *N. meningitidis*, *sodC* is not found in other *Neisseria* spp. Furthermore, there are no reports of meningococci lacking the *sodC* gene [26].

Antimicrobial resistance in *Neisseria meningitidis* remains a concern and varies across studies and countries. A global meta-analysis found low levels of antimicrobial resistance (1–3.4%) to ceftriaxone, cefotaxime, ciprofloxacin, and rifampin, but not to penicillin (27.2%) [27]. However, studies in Ethiopia have shown high levels of resistance to ciprofloxacin (50–60%), cotrimoxazole (62–100%), ceftriaxone (13–69.4%), and penicillin (95.8%) among asymptomatic carriers, with multidrug resistance in 14.3–60.4% of isolates [28,29,30]. These findings highlight the need for continued surveillance and appropriate antibiotic stewardship to effectively manage *N. meningitidis* infections.

The primary objective of this study was to develop and validate an in-house molecular method with improved sensitivity for the detection of *N. meningitidis* infection. To this end, an in-house *sodC*-targeted PCR assay was developed and validated in clinical (pharyngeal swabs) samples and culture-positive *N. meningitidis* isolates. The results from this study demonstrated that our in-house *sodC*-targeted PCR assay exhibited enhanced sensitivity compared to the *ctrA*-targeted assay in detecting *N. meningitidis* infections in carriers, supporting its adoption as a valuable diagnostic tool in resource-constrained microbiology laboratories. The ultimate goal of introducing diagnostic tests with improved sensitivity and specificity is to provide appropriate chemoprophylaxis and treatment strategies. Consequently, we set a secondary objective to assess the antimicrobial resistance profiles of *N. meningitidis* isolates from asymptomatic carriers between 2010 and 2012. These were also used to validate our in-house *sodC*-targeted PCR assay.

## 2. Materials and Methods

### 2.1. Ethical Consideration

This study was conducted on Ethiopian archived clinical (pharyngeal swabs) and culture samples collected from the MenAfriCar study, a multi-country cross-sectional survey conducted in seven African meningitis belt countries (Chad, Ethiopia, Ghana, Mali, Niger, Nigeria, and Senegal) in 2010, 2011, and 2012 [31]. An ethical waiver for the use of these archived isolates and clinical specimens was obtained from the All African Leprosy Rehabilitation Center (ALERT)/Armauer Hansen Research Institute (AHRI) Ethics Committee with approval number: PO-63-22.

### 2.2. Carriage Specimens

As mentioned above, we used stored samples collected from asymptomatic *Neisseria spp*. carriers as part of the MenAfriCar project [31]. The study samples consisted of randomly selected pharyngeal swabs (*n* = 137) and culture-positive *N. meningitidis* isolates (*n* = 49). These samples were stored at −80 °C in 1 mL STGG medium (containing skim milk, tryptisoya, glucose, and glycerol) at the AHRI Microbiology Laboratory. The 49 *N meningitidis* isolates were used to validate the in-house *sodC* gene-based PCR assay, while its performance comparison was evaluated in 137 clinical samples archived in 1 mL STGG.

### 2.3. Characterization and Confirmation of N. meningitidis Isolates

*N. meningitidis* was identified by microscopic examination and biochemical tests. Briefly, cryopreserved bacterial isolates were thawed at room temperature. A loopful of these samples was then sub-cultured onto fresh chocolate (CHO) and modified Thayer Martin (MTM) agar plates, which were supplemented with vancomycin, colistin, nystatin, trimethoprim (VCNT) (Oxoid^TM^ Hampshire, UK), and IsoVitaleX enrichment (Oxoid^TM^ Hampshire, UK). Colonies were further confirmed using Gram stain and oxidase tests. After confirming the presence of Gram-negative diplococci and oxidase-positive isolates, a pure colony was sub-cultured again on a blood agar plate (BAP) with 5–10% CO_2_ for 24 to 48 h to ensure the purity of the culture for further biochemical testing.

A carbohydrate utilization test (glucose, maltose, lactose, and sucrose) was performed using cystine trypticase agar (CTA) to differentiate *N. meningitidis* from Moraxella species and other nonpathogenic *Neisseria* species. Isolates that were Gram-negative diplococci, oxidase-positive, glucose fermenters, maltose fermenters, and non-fermenters of lactose and sucrose were confirmed as *N. meningitidis,* a total of 49 isolates.

### 2.4. Antimicrobial Susceptibility Testing (AST)

AST was performed on 49 isolates of *N. meningitidis*, all of which exhibited the standard characteristics of *N. meningitidis* colonies using Mueller−Hinton agar (MHA) with 5% sheep blood [32]. Briefly, 3–5 colonies from a blood agar plate were suspended, and the turbidity was adjusted to match a 0.5 McFarland standard. The surface of the MHA with 5% sheep blood agar was then thoroughly coated with the bacterial suspension using a sterile swab. After allowing the plate to dry for 3–5 min, antibiotic discs were evenly placed on the inoculated plate with sterile forceps. The plate was incubated in 5% CO_2_ at 37 °C for 24 h. The antibiotics selected for AST profiling were based on the Ministry of Health Ethiopia guidelines, fourth edition 2021 (https://www.slideshare.net/TesfayeWorkie/stg-2021pdf#1, accessed on 10 January 2025). The tested antibiotics included ceftriaxone (30 μg), meropenem (30 µg), cefepime (30 µg), trimethoprim−sulfamethoxazole (1.25/23.75 µg), ampicillin (10 µg), amoxicillin (10 µg), and ceftazidime (30 µg). The diameters of the inhibition zones around the discs were measured to the nearest millimeter using a graduated caliper, and these measurements were compared to standard charts to determine the susceptibility, intermediate resistance, or resistance of the bacteria to the tested antibiotics, according to the USA clinical and laboratory standard institute 2022 [33].

### 2.5. DNA Preparation and Quantification

DNA was isolated from 137 clinical samples (pharyngeal swabs) preserved in STGG broth, as well as from the 49 isolates. Briefly, samples stored in 1 mL of STGG broth at −80 °C were thawed at room temperature. Then, 200 μL of the sample was mixed with 200 μL of 1x TE buffer; in the case of the cultured sample, 3–5 colonies were picked up from the agar plate, then vortexed vigorously, and incubated in a water bath at 95 °C for 15 min. The tubes were then transferred to −20 °C for 10 min to release bacterial DNA through heat shock, followed by a 2 min incubation at room temperature. The tubes were centrifuged at 14,000 rpm for 5 min, and the genomic DNA remained in the upper aqueous phase (supernatant). The supernatant containing the DNA was transferred to a separate 2 mL sterile tube, and the DNA concentration was measured using a NanoDrop 2000/2000c spectrophotometer (Thermo Fisher Scientific, Waltham, MA, USA). The DNA was then stored at −80 °C until used as a template for the PCR assay.

### 2.6. In-House Development of sodC-Based PCR Assay

We selected the *sodC* gene as the target for developing a PCR assay aimed at detecting *N. meningitidis* in clinical samples. The *sodC* gene was chosen for several reasons: (i) it offers high specificity for detecting *N. meningitidis* as it is absent in other *Neisseria* species; (ii) the *sodC* gene in *N. meningitidis* encodes virulence factors and a periplasmic enzyme, making it less prone to antigenic variation due to selective pressure; and (iii) there are no known strains of meningococci that lack the *sodC* gene. These factors collectively suggest that a PCR assay based on the *sodC* gene can detect all *N. meningitidis* strains from various geographical regions without cross-reacting with other *Neisseria species* [19].

### 2.7. Primer Design

Primers (both forward and reverse) were designed using SnapGene Viewer (Table 1). The *sodC* sequences of *N. meningitidis* were sourced from Gene Bank (accession number >NZ_CP021520.1:999157-999717). The specificity of these primers to *N. meningitidis* was verified using the NCBI nucleotide BLAST tool.

### 2.8. sodC PCR Amplification Conditions

The *sodC* PCR reaction was performed in a 25 μL mixture, which included 0.625 μL of each primer (10 μM), 2.5 μL of dNTPs (2.5 μM), 0.5 μL of DNA polymerase, 2.5 μL of polymerase buffer with magnesium acetate (5 μL of 10x), 15.25 μL of nuclease-free water, and 3 μL of genomic DNA as the template. Each run incorporated both positive and negative controls. The optimized amplification protocol started with an initial denaturation at 95 °C for 3 min, followed by 35 cycles of denaturation at 94 °C for 10 s, annealing at 56 °C for 30 s, extension at 72 °C for 30 s, and a final extension at 72 °C for 10 min. The amplified product was then analyzed on a 2% agarose gel using 1x Tris-acetate EDTA (TAE) buffer and visualized with Bio-Rad UV scanning (a chemi PRO UV scanner, Hercules, CA, USA). The optimal conditions for the *sodC* PCR reaction and the most effective primer pairs were identified. This assay was then validated using the culture technique as the gold standard and then compared with the *ctrA* PCR assay.

### 2.9. ctrA Gene-Based PCR Assay

The ctrA gene-based PCR assay, which includes the paired oligonucleotide primer: forward primer (ctrA-F): 5′-GCTGCGGTAGGTGGTTCAA-3′ and reverse primer (ctrA-R): 5′-TTGTCGCGGATTTGCAACTA-3, was initially developed by Lansac et al. in 2000 [34], and later used at AHRI [35]. Briefly, the PCR reactions comprised 0.625 μL of both forward and reverse primers (10 μM each), 2.5 μL of dNTPs (2.5 μM), 0.5 μL of DNA polymerase, 2.5 μL of polymerase buffer with magnesium acetate (5 μL of 10x), 15.25 μL of nuclease-free water, and 3 μL of genomic DNA as the template, making a final volume of 25 μL. Each run included positive and negative controls. The amplification protocol was optimized with an initial denaturation at 95 °C for 3 min, followed by 30 s at 94 °C for denaturation, 45 s at 52 °C for annealing, and 1 min at 72 °C for extension, with a final extension at 72 °C for 10 min over 35 cycles. The amplified product was analyzed on a 2% agarose gel using 1x TAE buffer and visualized with Bio-Rad UV scanning (a chemi PRO UV scanner, Hercules, CA, USA). Representative gel images of the PCR products from both the *sodC* and *ctrA* genes, obtained from control strains ATCC (serogroup A: Z2491; W: A22; X: 860060; Y: 71/94) and suspected samples, are shown in Figure 1.

### 2.10. Data Management and Analysis

The data obtained were entered into Epi Info version 7 and exported into STATA version 14 for data cleaning and analysis. Results are presented in frequency tables and charts. Bivariate and multivariate logistic regression models were used to analyze data, and *p*-values less than 0.05 were considered statistically significant.

## 3. Results

### 3.1. Optimal Conditions for the sodC Gene-Based PCR Assay

To determine the optimal conditions for *sodC* gene-based PCR assay, a set of primer pairs targeting the *sodC* gene was optimized by using the temperature gradient of the annealing temperature (Figure 1) and concentration of oligonucleotide forward and reverse primer pairs (Table 1). The *sodC* gene target was amplified using a selected optimized concentration of pair of primers at optimized PCR conditions: initial denaturation at 95 °C for 3 min, followed by 35 cycles; denaturation at 94 °C for 10 s; annealing at 56 °C for 30 s; extension at 72 °C for 30 s; and a final extension at 72 °C for 10 min.

### 3.2. Performance Comparison Between sodC Gene-Based Detection of N. meningitidis and ctrA-Based Detection

To validate our in-house-developed *sodC* gene-based assay for detecting *N. meningitidis*, we utilized culture-positive *N. meningitidis* isolates and compared the results with those from the *ctrA*-based PCR detection method. Among the 49 DNA samples from culture-positive *N. meningitidis* isolates used for validation, the *sodC* gene-based PCR accurately identified all 49 culture-confirmed isolates. In contrast, the *ctrA* gene-based PCR detected only 33 of these isolates. This demonstrates a 100% concordance between the culture method and the *sodC* gene-based PCR assay (Table 2). After validating our in-house PCR assay targeting the *sodC* gene with culture-positive isolates, we assessed its concordance with the PCR assay targeting the *ctrA* gene using DNAs obtained from 137 pharyngeal swabs. The *sodC* gene-based method detected *N. meningitidis* DNA in 76.64% of the samples, whereas the *ctrA* gene-based method identified it in only 46.72% (Table 2).

### 3.3. Antimicrobial Susceptibility Pattern of N. meningitidis Isolates

The antimicrobial susceptibility patterns of the 49 *N. meningitidis* isolates are summarized in Table 3. Among these isolates, resistance was found in 43 (87.8%) to amoxicillin, 42 (83.7%) to ampicillin, 32 (65.3%) to trimethoprim–sulfamethoxazole, 22 (44.9%) to ceftazidime, and 18 (36.7%) to both ceftriaxone and meropenem. Additionally, seven isolates (15.2%) showed resistance to cefepime. On the other hand, a notable number of isolates were sensitive to cefepime (36 isolates, 73.5%), ceftriaxone and meropenem (31 isolates, 63.3%), and ceftazidime (26 isolates, 53.1%).

### 3.4. Antimicrobial Susceptibility Patterns of N. meningitidis Isolates by Age and Sex of Asymptomatic Carriers

The antimicrobial susceptibility patterns of the 49 *N. meningitidis* isolates, based on the age and sex of the asymptomatic carriers from whom they were isolated, are summarized in Table 4 and Table 5. No significant differences in antimicrobial susceptibility patterns were observed based on the sex and age of carriers, although slight variations in resistance percentiles were noted across different age groups for some antibiotics. For example, isolates from those aged 24–29 showed the lowest percentile of resistance to ampicillin and amoxicillin, while isolates from those aged 10–14 showed the highest percentile of resistance to these same antibiotics (Table 5).

## 4. Discussion

The results of this study offer important insights into the diagnostic techniques for the detection of *N. meningitidis*, particularly focusing on comparing PCR-targeted *sodC* and *ctrA* genes. The findings clearly show that, compared to the *ctrA* gene, the *sodC*-based PCR was a more sensitive and accurate method of detecting *N. meningitidis*.

The higher detection rate of *N. meningitidis* using PCR targeting the *sodC* gene highlights the potential of this method as an effective tool for the diagnosis of meningococcal disease [19,25]. In contrast, the lower detection rates of *N. meningitidis* using the *ctrA* gene methods suggest that this method may be less sensitive and reliable for detecting bacteria in clinical samples. This has important implications for clinical diagnosis, as timely and accurate detection of *N. meningitidis* is essential for initiating appropriate treatment and implementing public health measures to prevent the spread of the disease. In a similar study that tested pharyngeal swabs for the presence of *N. meningitidis* by PCR with *sodC* and *ctrA* as target genes, 75.8% (491/647) of clinical samples tested positive for the *sodC* gene [36].

Furthermore, *sodC*-based real-time PCR identified a higher detection rate of *N. meningitidis* isolates; 518 of 520 (99.6%) isolates of *N. meningitidis* were positive for *sodC*, whereas *ctrA* detection occurred only in 368 of 520 (70.8%) isolates. Similar to our report, gene-based PCR showed higher sensitivity to the *sodC* gene than to the *ctrA* gene [19].

Contrary to our finding, in another molecular assay for the detection of meningococci in normally sterile sites using *sodC* and *ctrA* genes, *sodC*-based RT-PCR was found to be 7.5% less sensitive than *ctrA*-based RT-PCR [37].

Resistance to trimethoprim–sulphamethoxazole was high among meningococcal isolates. The widespread resistance to this antibiotic is possibly due to the early introduction of sulphonamides [38]. In contrast, an investigation in children in Greece in 2004 revealed that ceftriaxone sensitivity was present in all isolates [39]. Additionally, we observed no significant differences in antimicrobial susceptibility patterns based on the sex and age of carriers, although slight variations in resistance percentiles were noted across different age groups for some antibiotics. This observation aligns with a previous study conducted in the Meskan and Mareko Districts, Gurage Zone, Ethiopia [28].

*N. meningitidis* is developing a resistance to antibiotics that are recommended for the management of meningococcal meningitis for epidemic response. These findings have important implications for clinical practice and public health. Healthcare providers should be aware of the drug susceptibility profile of *N. meningitidis* in their region and consider these factors when selecting antimicrobial therapy for patients with suspected or confirmed meningococcal infections. Additionally, public health efforts should focus on surveillance of antimicrobial resistance in *N. meningitidis* and the development of new treatment strategies to combat resistant strains. Furthermore, the ability of PCR targeting the *sodC* gene to detect a larger proportion of *N. meningitidis* isolates has implications for epidemiological studies and surveillance of meningococcal carriers [19,25]. Accurate and comprehensive surveillance data are essential for understanding the epidemiology and dynamics of *N. meningitidis* transmission, as well as for informing public health interventions such as vaccination strategies. The higher detection rate of *N. meningitidis* using PCR targeting the *sodC* gene may also be valuable for identifying asymptomatic carriers of the bacteria, which is critical for implementing targeted control measures to prevent outbreaks.

This study was not without its limitations. First, the sample size utilized for the performance evaluations was modest, though it was adequate. Second, while *sodC* is specific to *N. meningitidis*, this study did not assess the specificity of the in-house *sodC*-targeted PCR assay using a panel of DNAs from non-*N. meningitidis* isolates, including *N. sicca*, *N. gonorrhoeae*, *Streptococcus pneumoniae*, and *Haemophilus influenzae*.

Although the present study demonstrated that *SodC* gene-based assay is a promising method for the detection of non-groupable *N. meningitidis*, especially in carriage, its use for the diagnosis of meningococcal meningitis warrants caution because of the potential for false results and public health implications [19,37]. The *sodC* gene-based PCR assay has been shown to be less sensitive in sterile fluids (e.g., cerebrospinal fluid) than the *ctrA* gene-based assay [19]. In addition, it is critical to differentiate cases of bacterial meningitis (*N. meningitidis*, *S. pneumoniae*, *H. influenzae*, and *Streptococcus agalactiae*) to ensure effective treatment. However, some bacterial species, such as *H. influenzae*, have homologous *sodC* genes [40], which can lead to false-positive results. Therefore, combining the *sodC* gene-based assay with *ctrA* gene-based detection could help improve the accurate diagnosis of meningococcal meningitis, especially in cases of suspected bacterial meningitis, thereby improving patient management.

## 5. Conclusions

The *sodC* gene-based PCR assay was found to be superior in sensitivity in detecting *N. meningitidis* in carriage specimens compared with *ctrA* gene-based PCR. The observed high prevalence of antibiotic resistance warrants the need to continue to monitor antibiotic resistance that might influence treatment and for future implementation of chemoprophylaxis for carriers and those household members who have contacts with confirmed meningitis cases.

## Figures and Tables

**Figure 1 diagnostics-15-00637-f001:**
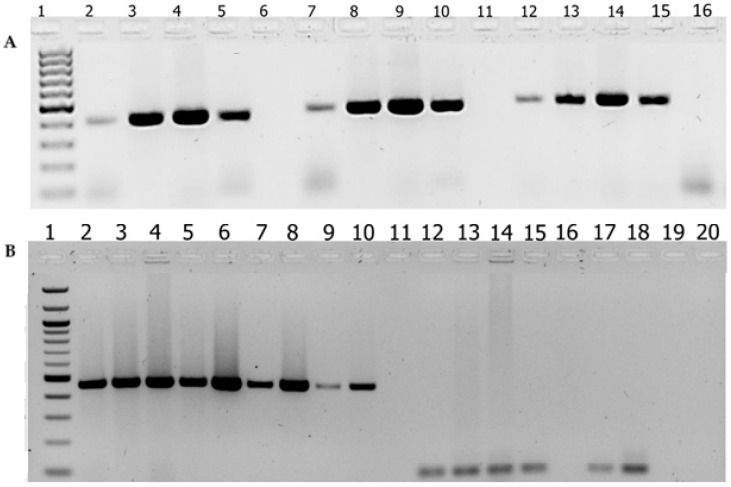
Agarose gel electrophoresis of PCR-amplified products using in-house *sodC*- and *ctrA*-targeted PCR assays. (**A**) Agarose gel electrophoresis of the PCR products obtained from optimization of PCR targeting the *sodC* gene. Lane 1 is 1 kb+ DNA ladder marker; Lanes 2, 7, and 12 are *N. meningitidis* ATCC control strain; and Lanes 3, 8, and 13, Lanes 4, 9, and 14, and Lanes 5, 10, and 15 are *N. meningitidis* isolates of serogroups X, Y, and W, respectively, amplified by three different versions of *sodC* primers. Lanes 6, 11, and 16 are negative controls. (**B**) Agarose gel electrophoresis of the PCR products obtained from PCR targeting the *sodC* gene and the *ctrA* gene. Lane 1, 1 kb+ DNA ladder marker; Lane 2, *N. meningitidis* ATCC control strain, Lanes 3–10, clinical samples amplified by primers targeting the *sodC*; Lane 11, negative control; and Lanes 12–20, clinical samples amplified by primers targeting the *ctrA* gene.

**Table 1 diagnostics-15-00637-t001:** List of oligonucleotide primers designed for the *N. meningitidis sodC* gene target.

S. No	Oligonucleotide	5′-3′ Nucleotide Sequences
1.	*sodC* Fw1-PCR	ATGAATATGAAAACCTTATTAGCACTAGCGGTTAGTGCAG
2.	*sodC* Fw14-48	CCTTATTAGCACTAGCGGTTAGTGCAGTATGTTC
3.	*sodC* Fw14-PCR	CCTTATTAGCACTAGCGGTTAG
4.	*sodC* Fw64	GCACACGAGCATAATACGATACCTAAAGGTGCTTC
5.	*sodC* Fw118	CAACTTGATCCAGCAAACGGTAACAAAGATGTGGG
6.	*sodC* Fw361	GCACACTTAGGTGATTTACCTGCATTAACTG
7.	*sodC* Rv478-PCR	GGATCATAATAGAGTGACCGCGAAC
8.	*sodC* Rv520-PCR	CAAGTGGAGCTGGATGATCGGAGTG
9.	*sodC* Rv561-PCR	TTATTTAATCACGCCACATGCCATACGTGG

**Table 2 diagnostics-15-00637-t002:** Performance comparison between two in-house PCR methods for detection of *Neisseria meningitidis* in carriage specimens: *N. meningitidis* isolates (*n* = 49) and pharyngeal swabs (*n* = 137).

In-House PCR Assay	Total	Positive	Negative
DNA from culture-confirmed isolates			
*sodC*	49	49	0
*ctrA*	49	33	16
DNA from clinical samples (pharyngeal swabs)			
*sodC*	137	105	32
*ctrA*	137	64	73

**Table 3 diagnostics-15-00637-t003:** Drug susceptibility profile of 49 *N. meningitidis* isolates.

Antimicrobials with Disk Content	Susceptibility Profile	No (49)	%
Ceftriaxone/CRO (30 µg)	I ^a^	-	-
	R ^b^	18	36.7
	S ^c^	31	63.3
Ampicillin (10 µg)	I	-	-
	R	42	83.7
	S	8	16.3
Amoxicillin (10 µg)	I	-	-
	R	43	87.8
	S	6	12.2
Trimethoprim–sulfamethoxazole1/SXT (1.25/23.75 µg)	I	6	12.2
	R	32	65.3
	S	11	22.5
Cefepime (30 µg)	I	3	11.3
	R	7	15.2
	S	36	73.5
Meropenem (30 µg)	I	-	-
	R	18	36.7
	S	31	63.3
Ceftazidime (30 µg)	I	1	2
	R	22	44.9
	S	26	53.1

^a^ I = intermediate. ^b^ R = resistance. ^c^ S = sensitivity.

**Table 4 diagnostics-15-00637-t004:** Bivariable logistic regression analysis for association between sex and antibiotic susceptibility pattern of 49 *N. meningitidis* isolates.

Antibiotics		Sex		OR (95% CI)	*p*-Value
		Male (*n* = 26)	Female (*n* = 23)		
Ceftriaxone	S	18	13	1.73 (0.54, 5.58)	0.36
	R	8	10		
Ampicillin	S	5	3	1.58 (0.33, 7.53)	0.56
	R	21	20		
Amoxicillin	S	3	3	0.87 (0.15, 4.80)	0.87
	R	23	20		
Trimethoprim–sulfamethoxazole/SXT	S	6	5	1.08 (0.28, 4.15)	0.91
	R	20	18		
Cefepime	S	20	19	0.70 (0.17, 2.88)	0.62
	R	6	4		
Meropenem	S	18	13	1.73 (0.54, 5.59)	0.36
	R	8	10		
Ceftazidime	S	15	11	1.49 (0.48, 4.60)	0.49
	R	11	12		

**Table 5 diagnostics-15-00637-t005:** Bivariable logistic regression analysis for the association between age and antibiotics susceptibility pattern of 49 *N. meningitidis* isolates.

Antibiotics		Age			OR (95% CI)	*p*-Value
		≤10	11–20	≥21		
Ceftriaxone	S	5	16	10	1.00 2.0 (0.54, 5.59) 1.28 (0.54, 5.59)	0.44 0.75
	R	4	10	4		
Ampicillin	S	2	4	2	1.00 0.64 (0.09, 4.24) 0.58 (0.06, 5.11)	0.64 0.63
	R	7	22	12		
Amoxicillin	S	1	3	2	1.00 1.04 (0.09, 11.52) 1.33 (0.10, 17.27)	0.97 0.82
	R	8	23	12		
Trimethoprim–sulfamethoxazole/SXT	S	2	6	3	1.00 0.95 (0.12, 7.23) 1.05 (0.17, 6.46)	0.96 0.95
	R	7	20	11		
Cefepime	S	8	19	12	1.00 0.75 (0.06, 9.72) 0.34 (0.03, 3.23)	0.82 0.34
	R	1	7	2		
Meropenem	S	5	16	10	1.00 2.0 (0.34, 11.54) 1.28 (0.27, 5.93)	0.44 0.75
	R	4	10	4		
Ceftazidime	S	3	14	9	1.00 3.6 (0.62, 21.03) 2.3 (0.47, 11.34)	0.15 0.23
	R	6	12	5		

## Data Availability

The data supporting the findings of this study are available upon reasonable request from the corresponding author.

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
