# Peer review of "Performance Comparison of Two In-House PCR Methods for Detecting Neisseria meningitidis in Asymptomatic Carriers and Antimicrobial Resistance Profiling"

_diagnostics, 2025, doi:10.3390/diagnostics15050637_

Round 1

Reviewer 1 Report

Comments and Suggestions for Authors

The authors presented interesting results of the comparison of two PCR methods in the diagnosis of N. meningitidis and the results of antimicrobial susceptibility testing. In my opinion, the title should be reformulated in such a way that it is clearly visible that two PCR methods were compered: in house metod for detecting the sodC gene and the method of detecting the ctrA gene in asymptomatic carriers.

The structured abstract is very well written. For a clearer presentation of the results and better readability, it would be good to divide the results into two groups:

1. results obtained by direct testing of nasopharyngeal swabs

2. test results after cultivation.

Line16: delete "was" and put "has been".

It is not clear which clinical samples authors used for asymptomatic bacteriology: nasopharyngeal swabs (line 32) or pharyngeal swabs (line 111).

In the introduction, it is necessary to add one or two sentences on the susceptibility and resistance of N. meningitidis in the world (not just refer to the references). Also, it's necessary to describe in more detail the aims of the study.

In the Materials and Methods, it's necessary to add the reference for the MenAfricar study. 

Line 118: "leftover" is an inappropriate word in this context. Please explain in detail how you stored the pharyngeal swabs (it's described letter in the text, but it should be explained here) and the number of clinical specimens used in the study (137?). This part (Study isolates and clinical samples) should be better explained.

Lines 130 - 142: Add the manufacturer's name for all culture media and state that N. meningitidis was identified based on microscopic examination and biochemical tests.

Add reference for CLSI (line 159).

Add reference for ctrA gene method (lines 201 - 215).

Add refrence for statistics (lines 222 - 226).

Line 225: sodC "italic"

Discussion and conclusions are well written, figures and tables are clearly presented.

Author Response

Dear reviewer, thank you very much for your invaluable comments and suggestions, they helped us a lot to improve our manuscript. All the changes we have made to address the issues you raised in the manuscript are highlighted in yellow.

Reviewer 2 Report

Comments and Suggestions for Authors

This study compared the performance of sodC versus ctrA methodologies in the detection of Neisseria Meningitides. The study also examined susceptibility to a number a number of antibiotics.

The introduction section will benefit for revision and rewriting. There are few repeats and duplication of information.

Too many tables were presented, the information of which could have been merely summarized in text in the results section.

Major revision is needed to the presentation. The significance of information provided is not clear, for instance, gendered different ion on antibiotic susceptibility. The authors need to justify inclusion of such data and their scientific merit. Particularly when the numbers are very small.

Validation of tests, in addition to assessment of detection and sensitivity, it should also address specificity. Were there any other species tested? Other non-Neisseria meningitides (known samples).

Although the study is well described, it would benefit from revision and consolidation of the vast number of tables and figures.

Author Response

Dear reviewer, thank you very much for your invaluable comments and suggestions, they helped us a lot to improve our manuscript. All the changes we have made to address the issues you raised in the manuscript are highlighted in yellow.

Comment 1 - The introduction section will benefit from revision and rewriting. There are a few repeats and duplication of information.

Response 1 - The induction was revised to address your and the other reviewer’s comments.

Comment 2 - Too many tables were presented, the information of which could have been merely summarized in text in the results section.

Response 2- As per your suggestion: Tables 1 and 3 deleted and information presented using Table 3 was described in the text; Figures 1 and 2 were merged into one figure and named figure 1; tables 4 and 5 related to the results on the performance evaluation of our assays were merged into one table, named Table 2.

Comment 3 - Major revision is needed to the presentation. The significance of information provided is not clear, for instance, gendered different on antibiotic susceptibility. The authors need to justify inclusion of such data and their scientific merit. Particularly when the numbers are very small.

Response 3– Dear reviewer we pointed out in the discussion section that even though we did not observe any significant differences in antimicrobial susceptibility patterns based on the sex and age of carriers, a slight variation in resistance percentiles were noted across different age groups for some antibiotics. There were also other reports with similar data. Thus, we thought there no harm in communicating such data with the scientific community.

Comment 4 - Validation of tests, in addition to assessment of detection and sensitivity, it should also address specificity. Were there any other species tested? Other non-Neisseria meningitides (known samples).

Response 4 - This is a very valid point. We thank the reviewer for pointing out this. Unfortunately, we do not have access to non-Neisseria meningitidis isolates. Future studies should include testing of DNA obtained from non-meningococcal isolates to evaluate the specificity of our in-house sodC-targeted PCR assay for the detection of N. meningitidis. We acknowledge this as a main limitation of our study and include it in the limitations se

Round 2

Reviewer 2 Report

Comments and Suggestions for Authors

Much improved and recommendations addressed. Stilled concerned about the lack of specificity studies, but the authors cited the limitation in the discussion. It would be helpful if the authors would expand an additional paragraph on the likelihood for impact specificity of their reagents and on the clinical impact or its likelihood if there were significant assay specificity drawback. E.g. What would be an acceptable testing algorithm to rule out possible interference. 

Author Response

Comment -1. Much improved and recommendations addressed. Stilled was concerned about the lack of specificity studies, but the authors cited the limitation in the discussion. It would be helpful if the authors would expand an additional paragraph on the likelihood of impact specificity of their reagents and the clinical impact or its likelihood if there were significant assay specificity drawbacks. E.g. What would be an acceptable testing algorithm to rule out possible interference.

Response -1. Dear reviewer, we thank you for the additional valuable comment. We agreed on the importance of adding a paragraph. Accordingly, we added the following paragraph regarding the cautious interpretation of results using only sodC gene-targeted detection of Nm in meningococcal meningitis cases.

“Although the present study demonstrates that SodC gene-based assay is a promising method for the detection of non-groupable N. meningitidis, especially in carriage, its use for the diagnosis of meningococcal meningitis warrants caution because of the potential for false results and public health implications [17, 35]. The sodC gene-based PCR assay is less sensitive in sterile fluids (e.g., cerebrospinal fluid) than the ctrA-based assay [17]. In addition, it is critical to differentiate cases of bacterial meningitis (N. meningitidis, S. pneumoniae, H. influenzae, and Streptococcus agalactiae) to ensure effective treatment. However, some bacterial species, such as H. influenzae, have homologous sodC genes [38], which can lead to false-positive results. Therefore, combining the sodC gene-based assay with ctrA gene-based detection could help improve the accurate diagnosis of meningococcal meningitis, especially in cases of suspected bacterial meningitis, thereby improving patient management. “